# Cell-Free RNA from Plasma in Patients with Neuroblastoma: Exploring the Technical and Clinical Potential

**DOI:** 10.3390/cancers15072108

**Published:** 2023-03-31

**Authors:** Nathalie S. M. Lak, Anne Seijger, Lieke M. J. van Zogchel, Nina U. Gelineau, Ahmad Javadi, Lily Zappeij-Kannegieter, Laura Bongiovanni, Anneloes Andriessen, Janine Stutterheim, C. Ellen van der Schoot, Alain de Bruin, Godelieve A. M. Tytgat

**Affiliations:** 1Princess Máxima Center for Pediatric Oncology, 3584 CS Utrecht, The Netherlands; 2Department of Experimental Immunohematology, Sanquin Research, 1066 CX Amsterdam, The Netherlands; 3Department of Biomolecular Health Sciences, Faculty of Veterinary Medicine, Utrecht University, 3584 CL Utrecht, The Netherlands; 4Department of Veterinary Medicine, University of Teramo, 64100 Teramo, Italy

**Keywords:** pediatric, solid tumors, neuroblastoma, cell-free RNA, liquid biopsies, extracellular vesicles

## Abstract

**Simple Summary:**

Neuroblastoma mostly affects young children and despite intensive treatment, many children die of progressive disease. It remains challenging to identify those patients at risk. Analyzing blood, as liquid biopsies, is not invasive and can help to identify these patients. We studied whether RNA molecules can be detected in these liquid biopsies. In blood plasma, RNA can be free-floating or packed in small particles, ‘extracellular vesicles’. We present a workflow to analyze this cell-free RNA from small volumes of blood plasma of children with neuroblastoma. We have used neuroblastoma-specific markers and markers involved in cell proliferation. These latter genes can be upregulated in many different tumor types. We demonstrate that both types of markers have a higher expression in patients with metastatic disease, compared to healthy controls and patients with localized disease. These findings are essential for future studies on cell-free RNA, hopefully leading to improved survival for these patients.

**Abstract:**

Neuroblastoma affects mostly young children, bearing a high morbidity and mortality. Liquid biopsies, e.g., molecular analysis of circulating tumor-derived nucleic acids in blood, offer a minimally invasive diagnostic modality. Cell-free RNA (cfRNA) is released by all cells, especially cancer. It circulates in blood packed in extracellular vesicles (EV) or attached to proteins. We studied the feasibility of analyzing cfRNA and EV, isolated by size exclusion chromatography (SEC), from platelet-poor plasma from healthy controls (*n* = 40) and neuroblastoma patients with localized (*n* = 10) and metastatic disease (*n* = 30). The mRNA content was determined using several multiplex droplet digital PCR (ddPCR) assays for a neuroblastoma-specific gene panel (*PHOX2B*, *TH*, *CHRNA3*) and a cell cycle regulation panel (*E2F1*, *CDC6*, *ATAD2*, *H2AFZ*, *MCM2*, *DHFR*). We applied corrections for the presence of platelets. We demonstrated that neuroblastoma-specific markers were present in plasma from 14/30 patients with metastatic disease and not in healthy controls and patients with localized disease. Most cell cycle markers had a higher expression in patients. The mRNA markers were mostly present in the EV-enriched SEC fractions. In conclusion, cfRNA can be isolated from plasma and EV and analyzed using multiplex ddPCR. cfRNA is an interesting novel liquid biopsy-based target to explore further.

## 1. Introduction

Neuroblastoma is the most common extracranial solid tumor in children [1]. Most patients present with disseminated disease which requires intensive treatment, consisting of chemotherapy, surgery and immunotherapy [1]. Still, more than half of patients suffer from refractory disease or relapse, which is associated with low survival [2,3]. At initial diagnosis and during the first courses of chemotherapy, it is hard to identify patients with treatment-resistant disease or at risk for relapse. Currently, response evaluation depends on imaging which often demands general anesthesia in these young patients. Liquid biopsy-based monitoring might decrease the number of diagnostic procedures and potentially even improve sensitivity of response monitoring [4,5].

The presence of neuroblastoma-specific mRNA in the cellular compartment of blood and bone marrow, such as *PHOX2B*, *TH* and *CHRNA3*, has been shown to correlate with outcome, enabling response monitoring in patients with high-risk disease [6,7,8,9]. Additionally, several targets in cell-free DNA (cfDNA) from plasma have been described to track therapy response, disappearing as tumor burden decreases and re-appearing as the disease relapses [10,11,12,13]. However, the presence of tumor-specific mRNA is often attributed to circulating tumor cells, which are not always present in every stage of the disease. cfDNA targets such as mutations in the *ALK* gene, amplification of *MYCN* or hypermethylation of the tumor suppressor gene *RASSF1A* (*RASSF1A*-M) are only applicable in patients with high-risk disease [10,12]. Therefore, apart from cfDNA, other liquid biopsy-based biomarkers in the plasma compartment deserve to be investigated. cfDNA is often shed through apoptosis or necrosis [14], whereas RNA is also actively secreted by living cells [15], presumably presenting a more comprehensive perspective on the ongoing disease [16]. Due to the presence of RNases in plasma [17], RNA has historically been considered unstable in plasma and therefore cfRNA not suitable for biomarker studies. However, in recent years, it has been discovered that plasma contains several types of RNA, which are mostly protected from degradation through their association with extracellular vesicles (EVs) or protein aggregates [16,18,19,20,21]. Furthermore, platelets contain RNA which also bears biomarker potential [22,23,24].

In the field of neuroblastoma, Morini et al. identified a panel of miRNA and showed that upregulation of these miRNA in plasma after induction therapy was associated with better chemotherapy response [25]. Ma et al. identified a single miRNA (miR199a-3p) which was upregulated in plasma from patients with neuroblastoma in all risk groups [26]. Recently, Matthew et al. have performed an impressive sequencing effort and characterized cell-free mRNA from plasma from both healthy controls and adults with lung and breast cancer [27]. They demonstrated that cfRNA expression profiles in patients differed from healthy controls, and they were able to identify tumor tissue-specific signatures. So far, similar sequencing studies in neuroblastoma have not been performed.

Another example of the possibilities of cell-free mRNA from plasma in cancer comes from studies in canines. Duplication of genomic DNA and distribution amongst the new daughter cells is a normal process in healthy cells. This process, named ‘cell cycle’, consists of well-defined phases, all guarded by checkpoints and their respective regulatory genes [28]. Tumor cells are highly proliferative due to dysregulation of the cell cycle [29]. A pivotal gene for the progression of the G2 phase to the S phase is *E2F1* [29,30]. In canines, Bongiovanni et al. identified several genes within the *E2F1* pathway to be overexpressed in tissue from canine melanomas, amongst them *E2F1*, *DHFR*, *CDC6*, *ATAD2*, *MCM2* and *H2AFZ* [31]. Subsequently, Andriessen et al. reported that *CDC6*, *DHFR*, *H2AFZ* and *ATAD2* transcripts were present in plasma of canines with malignancies and that these genes were mainly associated with EV [32]. Cell cycle dysregulation is an important feature of the pathogenesis of neuroblastoma [1,33], and we therefore postulated that transcripts of cell cycle proteins might potentially serve as novel biomarkers for this disease.

In this study, we explore the feasibility of detecting and studying cfRNA in plasma from patients with neuroblastoma by studying both a neuroblastoma-specific and a cell cycle panel for use on cell-free mRNA from plasma in patients with neuroblastoma. We report on the development of several multiplex panels for droplet digital PCR (ddPCR) and investigate whether these mRNA targets from plasma are associated with EVs. Finally, we describe technical challenges arising from the study of cfRNA from plasma.

## 2. Methods

### 2.1. Patients and Samples

Peripheral blood samples from neuroblastoma patients were collected within the Minimal Residual Disease study of the DCOG high-risk protocol, approved by the ethical committee of the Academic Medical Center, Amsterdam, The Netherlands (MEC07/219#08.17.0836). Samples from patients with International Neuroblastoma Staging System (INSS) stage 1 (localized disease that can be fully resected) and INSS stage 4 (metastatic disease) were included. Peripheral blood was collected in EDTA tubes (Becton-Dickinson, Franklin Lakes, NJ, USA) and processed within 24 h. Plasma was obtained by centrifugating blood samples at 1375× *g* for 10 min and stored at −20 °C until further processing. For controls, blood was collected from healthy adult volunteers and prepared similar to patients’ samples, including storage at −20 °C.

### 2.2. Preparation of Platelets from Peripheral Blood

Peripheral blood was collected in EDTA tubes (Becton-Dickinson) and processed within 2 h. First, platelet-rich plasma was obtained by centrifugation at 235× *g* for 15 min. The supernatant was collected and 10% anticoagulant citrate dextrose, solution A (ACD-A, Terumo, Japan)) was added and centrifuged at 16,873× *g* for 4 min to pellet the platelets. Leukocyte and platelet counts were measured with the Sysmex XN1000 Hematology analyzer (Sysmex, Kobe, Japan) according to manufacturer’s protocol.

### 2.3. Isolation of Cell-Free RNA and cDNA Synthesis

RNA was isolated from 200 µL of plasma, unless otherwise specified, with the miRNeasy micro serum/plasma kit (Qiagen, Germantown, TN, USA) following manufacturer’s protocol. RNA was eluted in 12 µL of H2O and subsequently used for cDNA synthesis with the High Capacity RNA-to-cDNA kit (Thermo Fisher, Waltham, MA, USA).

### 2.4. Design and Optimization of the Multiplex ddPCR Assays

For the detection of *PHOX2B*, *TH* and *CHRNA3*, the same primers and probes as previously described for RT-qPCR were used [6,9,34]. As potential cfRNA reference genes, *GUSB* and *B2M* were included, as previously described for RT-qPCR [35]. Genes involved in the *E2F1* pathway were *CDC6*, *ATAD2*, *DHFR*, *H2AFZ* and *MCM2*. To quantify the presence of platelets in the plasma, we applied an assay for platelet-specific *ITG3B*, designed to amplify and detect both polymorphic alleles (*HPA-1A* and *HPA-1B*) of this gene [29]. ddPCR assays were designed using Primer3Plus (www.primer3plus.com (accessed on 1 February 2021)). All sequences are shown in Appendix A. The QX200™ Droplet Generator (Bio Rad, Hercules, CA, USA) or QX200™ Automated Droplet Generator (Bio Rad) were used for droplet generation. Thermal cycling was performed using the C1000 Touch Thermal Cycler (Bio Rad) with the following program: 95 °C for 10 min; 40 cycles of 94 °C for 30 s, annealing temperature variable per assay for 1 min; 98 °C for 10 min; 4 °C hold. Following PCR, droplets were read and quantified using the QX200 Droplet reader (Bio Rad). Assays were optimized using RNA isolated from the neuroblastoma cell line IMR32 or RNA isolated from healthy platelets. All patient samples were tested in duplicate and ‘no template controls’ were included with every assay. ddPCR assay analyses were done in QX Manager 1.2 Standard Edition software (Bio Rad), except if indicated, then analyzed in Quantasoft 1.7.4 software (Bio Rad). Results are represented in copies/mL plasma, unless otherwise specified.

### 2.5. Isolation of Cell-Free DNA and ddPCR Assays

cfDNA was isolated using the Quick cfDNA Serum & Plasma kit (Zymo Research, Irvine, CA, USA). The methylation-sensitive restriction enzyme-based ddPCR for methylated tumor suppressor gene *RASSF1A* (*RASSF1A*-M) and *ACTB* was performed as described previously [10].

### 2.6. Isolation of EVs from Plasma and Electron Microscopy on EVs

EVs from plasma were isolated from 500 µL plasma by size exclusion chromatography (SEC) columns (qEV Original 70 nm from Izon Science, Christchurch, New Zealand) according to manufacturer’s protocol. SEC fractions 6 to 20 were collected. Electron microscopy was performed as reported previously [30].

### 2.7. Western Blot

Protein content of each SEC fraction was measured by micro BCA protein assay (Thermo Fisher Scientific), and input from the separate SEC fractions was adjusted accordingly to obtain equal loading of every SEC fraction onto the 4–12% SDS PAGE gel (Bio Rad). Protein concentration was eventually determined by a precipitation assay with trichloroacetic acid (Sigma, Kanagawa, Japan). After transfer to a nitrocellulose membrane (Bio Rad), the membrane was cut into two parts to allow for staining for different targets simultaneously. The membrane was blocked with PBS containing 5% (*w*/*v*) bovine serum albumin and then incubated with CD9 (Santa Cruz Biotechnology, Dallas, TX, USA, sc52519, 1:1000) and CD63 (BD Biosciences, San Jose, CA, USA, 556019, 1:1000). Antibody binding was visualized with anti-mouse IgG coupled to horse radish peroxidase at a 1:5000 dilution. Subsequently, the membranes were stripped by incubating with 1% NaN3 for an hour, and after blocking, incubated with CD81 (Santa Cruz Biotechnology, Santa Cruz, Dallas, TX, USA, SC9158, 1:1000) and TSG101 (Sigma, St. Louis, MI, USA, T5701, 1:1000).

### 2.8. Statistical Analysis

Statistical analyses were performed using SPSS version 23. Venn diagrams were generated using Lucid chart (www.lucidchart.com (accessed on 18 February 2022)). All other figures were generated using GraphPad Prism version 8. Continuous variables were analyzed using the non-parametric Mann–Whitney U test; differences were considered significant at *p* < 0.05.

## 3. Results

### 3.1. Neuroblastoma-Specific mRNA Is Present in Plasma

To study cfRNA in limited volume samples of pediatric patients with neuroblastoma, we first designed and optimized a multiplex ddPCR which included the neuroblastoma-specific targets *PHOX2B*, *CHRNA3* and *TH*, and *GUSB* as a reference gene (Appendix A). In 40 healthy controls, there were no transcripts of *PHOX2B*, *TH* or *CHRNA3* detected, whereas in all donors, *GUSB* transcripts could be demonstrated (mean 96 copies/mL plasma, range 36–238 copies/mL) (Appendix A).

We tested the neuroblastoma-specific multiplex ddPCR panel in a first cohort, consisting of 38 samples from 22 patients with neuroblastoma, which were collected at different timepoints during treatment (patient characteristics and outcome in Appendix A). In these 38 samples, only 24 samples were positive for *GUSB* and at lower concentrations (mean of positive samples 14.9 copies/mL plasma (range 2.0–127 copies/mL)). In the 24 samples positive for *GUSB*, 2 samples were positive for *PHOX2B* and *GUSB* (1 at initial diagnosis (2.1 and 2.1 copies/mL, respectively) and 1 at relapse (11 and 127 copies/mL), Appendix A). No samples were positive for *TH* or *CHRNA3*. As it is known that freeze–thaw cycles can affect cfRNA quality [27], we hypothesized that the cfRNA in the samples from these archived samples might be degenerated. Unfortunately, no RNA or plasma was left for analysis of RNA quality through another modality, e.g., Bioanalyzer.

To overcome this problem, we subsequently used only pre-treatment plasma samples that had not been thawed before, 10 samples from patients with INSS stage 1 (localized disease) and 30 INSS stage 4 neuroblastoma patients (metastatic disease), to form a second cohort. Patient characteristics and outcomes are shown in Appendix A. Results for the neuroblastoma-specific markers are shown in Figure 1 and Appendix A. In all 40 neuroblastoma samples, *GUSB* was detectable; for patients with localized disease, the mean was 38.2 copies/mL plasma (range 2.3–95 copies/mL plasma) and metastatic disease, 53 copies/mL plasma (range 10–220 copies/mL plasma). In none of the samples of patients with localized disease were *PHOX2B*, *TH* and *CHRNA3* detected. In contrast, in 14/30 samples of patients with metastatic disease, *PHOX2B* (*n* = 13, 9.2 copies/mL, range 0.4–47 copies/mL) and/or *CHRNA3* (*n* = 4, mean 5.4 copies/mL, range 2.1–11 copies/mL) was detected. No samples were positive for *TH*. In the samples with at least one marker positive, 10/14 (71%) suffered from an event vs. 11/16 (69%) in the negative samples.

### 3.2. Cell Cycle Genes in Plasma and Correction for the Presence of Platelets

Next, we investigated the presence of transcripts of cell cycle genes in cfRNA (Appendix A for the cell cycle panel ddPCR assays). We found that platelets also contain these transcripts (Appendix A for expression in platelets). Since plasma was isolated by centrifugation at 1375× *g* for 10 min, contaminating platelets might have been present in the plasma thereby affecting the analysis. Indeed, in EDTA blood from four healthy controls that were treated similarly as our plasma samples, using the Sysmex system, we measured that after the centrifugation step, 25–50% of the platelets were still present in plasma (Appendix A). As platelet counts can vary between patients and healthy controls, the cfRNA was corrected for the presence of platelet-specific RNA using the platelet-specific *ITGB3* ddPCR. The ratio between *ITGB3* expression and the different cell cycle gene transcripts was stable between donors, which enabled a correction co-efficient for each marker, as indicated in Appendix A.

### 3.3. Cell Cycle Genes in Plasma from Patients at Diagnosis

We measured the expression of the six cell cycle genes (*CDC6*, *ATAD2*, *E2F1*, *H2AFZ*, *MCM2* and *DHFR*), the two potential references genes (*GUSB* and *B2M*) and the platelet-specific marker *ITGB3* in 200 µL of plasma from 20 healthy controls (Appendix A). We then proceeded to measure these genes in our cohort of 40 patients. After correcting for platelets, CDC6, *ATAD2*, *DHFR*, *E2F1*, *H2AFZ*, *GUSB* and *B2M* were significantly higher in patients with localized disease than in healthy controls. *CDC6*, *DHFR*, *E2F1*, *H2AFZ*, *MCM2*, *GUSB* and *B2M* were significantly higher in patients with metastatic disease than in healthy controls, and *CDC6*, *DHFR* and *E2F1* were significantly higher in metastatic patients than in localized patients (Figure 2 and Appendix A).

We hypothesized that these cell cycle panels could assist in differentiating patients from healthy controls and could possibly differentiate between low- and high-risk disease. For this purpose, we determined the background expression of the cell cycle markers in 20 healthy plasma samples and set a threshold for positivity (Appendix A). When applying the thresholds for positivity per marker (after correcting for platelets), none of the 10 patients with localized disease were positive, whereas 14 out of 30 patients with metastatic disease had markers that were above the threshold (Appendix A). All of these 14 patients were positive for DHFR, and only 3 patients were also positive for *MCM2* in combination with *CDC6* (*n* = 2) and *H2AFZ* (*n* = 1). All three patients suffered from relapse. Eleven other patients were only positive for *DHFR*. A total of 7/11 suffered from relapse or refractory disease, and 4 eventually died from the disease. In this small cohort, we observed that, when correcting for platelets and background expression, *DHFR* was elevated in 14/30 patients with metastatic disease at diagnosis. When compared with the neuroblastoma genes, 7/14 *DHFR*-positive patients were also positive for *PHOX2B* and/or *CHRNA3*.

### 3.4. cfRNA during Treatment

To explore the potential of cfRNA measurements to monitor residual disease during treatment, we measured 66 samples drawn during treatment from 11 patients with metastatic disease (Appendix A). Patients were chosen according to their clinical outcome and availability of follow-up samples. All neuroblastoma-specific markers were negative in all follow-up samples, except for one sample at the first course of first-line chemotherapy in patient NBL 2187. This sample was positive for *CHRNA3* (2.0 copies/mL plasma). We also measured the cell cycle markers in the sequential samples. Many neuroblastoma patients suffer from bone marrow depression due to toxicity of chemotherapy during treatment, which results in low platelet counts. Since we do not know how this affects the RNA content of platelets and if the cell cycle/*ITGB3* ratios are affected, we decided not to use the *ITGB3*-corrected ratios for these samples but to only use the absolute number of copies present in the samples. Appendix A displays the course of the markers for all 11 patients, sorted per clinical outcome. From seven patients, at least three samples during the first line of therapy were available, and from three patients, two samples during the first line of therapy. In all patients, *B2M* always had the highest expression throughout the entire treatment. The other transcripts varied greatly per patient. No marker showed an evident increase or decrease in expression in patients with good vs. poor clinical outcome. Therefore, it is impossible to draw a conclusion on the level of specific markers in relation to clinical outcome in this small cohort, considering the variation in sampled time points, the unknown platelet counts and variation in expression levels between the different patients.

### 3.5. The mRNA in Plasma Is Concentrated in EV-Enriched SEC Fractions

Subsequently, we investigated in which compartment the neuroblastoma-derived transcripts were present in a patient with metastatic disease by size exclusion chromatography (SEC) on 500 µL of plasma, yielding SEC fractions of 500 µL each. The mRNA markers were tested in parallel to cfDNA using the reference gene *ACTB* and tumor-specific *RASSF1A*-M. The presence of EV was confirmed on western blot by the presence of EV-enriched proteins CD9, CD63, CD81 and TSG101 in SEC fractions 7 to 10 isolated from a healthy control and a patient with metastatic disease (Appendix A). Electron microscopy on SEC fractions from the same patient also confirmed the presence of EV in fractions 7 to 10 (Appendix A), whereas the higher fractions contained aggregated proteins. ddPCR of the RNA markers (both neuroblastoma-specific and cell cycle) and DNA markers from 200 µL of each SEC fraction from two other patients with metastatic disease (one (NBL2196) being *PHOX2B*-positive in unfractionated plasma and one *PHOX2B*-negative (NBL2187)) showed that the mRNA markers were mostly present in the EV-enriched fractions, whereas the DNA targets were mostly present in the higher, protein-enriched fractions (Figure 3A,B,D,E, Appendix A).

Please note that due to a high concentration of *B2M*, *H2AFZ* and *HPA1A/B*, the insert in Figure 3C,F displays an adjusted *y*-axis without these markers to show the concentration of the other markers.

The presence of mRNA (neuroblastoma-specific and cell cycle markers) in the EV fractions was confirmed in a subsequent experiment with another patient (NBL2177) and a healthy control in which the input in the cDNA reaction was increased 2.5-fold by using the complete 500 µL of SEC fractions. The results are shown in Figure 4 and Appendix A. Overall, the sum of the positive droplets from all SEC fractions corresponds well to what is found in 500 µL of whole plasma. Unexpectedly, *DHFR* is increased in the higher, protein-enriched fraction in the patient sample and is even higher than *B2M*.

## 4. Discussion

This study addressed the potential for cfRNA analysis from small volumes of plasma using multiplex ddPCR assays and EV enrichment. Within this first exploratory study, we did not aim to draw conclusions on added value to current clinical practice. However, we demonstrated that in patients with neuroblastoma, neuroblastoma-specific cfRNA is only present in patients with metastatic disease and that this RNA is associated with EV. Even with low volumes of plasma, neuroblastoma-specific and quantifiable signals can be obtained when using multiplex ddPCR assays. In this small patient cohort, no correlation with outcome of the disease was observed.

However, we did identify several challenges that are essential to further studies on cfRNA in neuroblastoma. Firstly, pre-analytical variables concerning the preparation and storage of the plasma are critical [20]. The plasma samples we used were prepared within 24 h after collection and then stored at −20 °C. In our first cohort, we used plasma samples that had gone through several cycles of thawing and freezing, and from many of these plasmas, no intact mRNA could be isolated, in contrast to plasma from healthy controls that was only thawed once for the cfRNA isolation. This observation that one freeze–thaw cycle does not affect RNA content is also described by Matthew et al. [27].

The plasma preparation protocol is an equally important consideration if one aims to study cfRNA and the transcripts of interest are also expressed in platelets. In this cohort, a one-step centrifugation protocol was applied to obtain platelet-poor plasma (as we confirmed). Since cell cycle genes are expressed in healthy platelets, the quantitative data had to be corrected for the presence of the variable number of platelets. As we showed that the ratio between our transcripts of interest and the platelet-specific transcript *ITGB3* was similar between platelets of different healthy donors, it was possible to correct our data. For future studies, the use of platelet-free plasma might be preferable. Platelet-free plasma was not collected for our cohort but is worth considering in future prospective studies on cfRNA.

Considering the lack of literature on reference genes for cfRNA, we pragmatically included *GUSB*, which is regularly used as a reference gene for the cellular compartment of peripheral blood for patients with neuroblastoma [9]. In addition, we included *B2M* as a reference gene [35] as it has been described that *B2M* is one of the genes highly expressed in plasma, although this finding might partly be caused by its high expression in platelets, as is shown in our data and known from the literature [23,27]. *B2M* is also an interesting gene specifically for neuroblastoma. It is part of the major histocompatibility complex (MHC), and neuroblastoma cells downregulate MHC proteins, probably in order to evade the immune system [36,37]. Our data suggest that this downregulation is not mediated by specific expulsion of *B2M* mRNA from the neuroblastoma cells.

The literature on cfRNA analysis in neuroblastoma patients is scarce. Only a single report by Corrias et al. reports on the analysis of cfRNA by RT-qPCR in neuroblastoma patients, stages 1 to 4, using *TH* as the only neuroblastoma-specific marker and several reference genes, including *B2M* [38]. This study aimed to investigate whether the analysis of cfRNA was useful in monitoring disease status, as compared to analysis with the same markers in whole blood. In this study, 6 out of 47 samples were positive for TH (1/4 patients with stage 3 disease at diagnosis, 0/15 patients with stage 4 at diagnosis, 1/13 patients with stage 4 during treatment, and 4/15 patients at relapse). In our study, we increased the number of neuroblastoma-specific genes, and by using the ddPCR instead of RT-qPCR, we could increase sensitivity and were able to precisely quantitate the number of transcripts. Interestingly, in our study, no samples were positive for *TH*, but almost half of the stage 4 samples were positive for *PHOX2B* and/or *CHRNA3* at diagnosis. In contrast, none of the 10 diagnostic samples from patients with localized disease tested positive, strongly suggesting that the presence of cfRNA is related to the stage of the disease. Corrias et al. conclude that the analysis of cfRNA with *TH* was not superior for monitoring of treatment response compared to RNA analysis from blood cells. Although our discovery study was not aimed or powered to study this question, our study also found that in only one of the 66 samples obtained during treatment could tumor-specific cfRNA be demonstrated. In addition, we did not find a prognostic difference for patients testing positive for the neuroblastoma-specific cfRNA at diagnosis compared to the negative patients (71% vs. 69%, respectively).

It is known that cell cycle genes belonging to the *E2F1* pathway can be highly expressed in malignancies. For neuroblastoma, it is known that *MYCN* amplification is a feature of aggressive disease [1] and that this in turn can upregulate *E2F1* and *MCM2* [39,40,41]. *CDC6*, one of the genes of the *E2F1* pathway, is also described as an important player in cell proliferation and cell death in neuroblastoma cells, as illustrated by knockdown experiments by Feng et al. [42]. Recently, Andriessen et al. demonstrated that *CDC6* was significantly elevated in plasma from canine patients with malignancies compared to healthy controls [32]. This prompted us to study cell cycle genes in our cohort as well, aiming to overcome the low expression of the neuroblastoma-specific markers in cfRNA. Although *CDC6* as well as *E2F1* and *DHFR* were, after correcting for platelet presence, significantly higher in patients with metastatic disease compared to patients with localized disease and healthy controls, these markers cannot be easily used to discriminate between healthy individuals and neuroblastoma patients at the individual level. Only two patients had increased levels of *CDC6* and *MCM2* after correcting for background expression; one patient had increased levels of *MCM2* and other genes were not increased, except for *DHFR*. This latter gene was elevated in 14/30 patients with metastatic disease. *DHFR* plays a role in the cell cycle as an enzyme in the folate biosynthesis pathway and is thereby essential for cell proliferation. Inhibition of *DHFR* has been used historically in antimicrobial agents (e.g., trimethoprim), rheumatoid disease and cancer (e.g., methotrexate) [43,44]. Our findings on elevated *DHFR* at diagnosis in neuroblastoma patients with metastatic disease support its crucial role, corresponding to a high proliferation rate in metastatic disease. However, if the level of DHFR would have a linear association with tumor burden, we would expect this to be reflected in the longitudinal samples. But this was not observed in our cohort.

Indeed, in the longitudinal samples, all cell cycle markers had a high variation in expression between patients and within individual patients, whereas B2M was consistently high. Sample collection was not consistent in our cohort beyond the induction treatment phase, which further complicated speculations on their potential as markers for early treatment failure or relapse detection. This underlines that further studies with standardized sampling are essential for future cfRNA research. In addition, studies broadening the perspective on mRNA markers specifically for cfRNA research are necessary since current RNA markers are mostly based on the cellular compartment of blood and might not be suited for cfRNA. Specifically for patients with neuroblastoma, a study including RNA sequencing of the transcriptome of the cell-free compartment at different timepoints could improve understanding of this field immensely and help identify cfRNA markers with diagnostic and prognostic value.

We confirmed that most of the mRNA markers are concentrated in the EV-enriched fractions [18,21,32]. However, unexpectedly, *DHFR* was found to be mostly present in the higher SEC fractions. This could be due to elution of smaller EV in later fractions or also to packaging of mRNA into protein aggregates; both hypotheses are supported by the literature [21,45,46,47,48] and the presence of *B2M* and *GUSB* in these SEC fractions. Further studies using RNase and proteinase on the different SEC fractions could elucidate further if mRNA is truly packaged within EV or only associated with EV. The same approach is possible for cfDNA using DNase, since we mostly demonstrate the presence of cfDNA in the protein-enriched fractions. The literature is still conflicting on this subject [49,50], and it is not inconceivable that EV cargo could even differ per disease. Furthermore, the method chosen for EV enrichment heavily affects the result of the downstream analysis, and after SEC, the presence of similar-sized lipoprotein particles in the EV-enriched fractions might also result in less pure EV preparations [51,52].

Considering a possible implementation in clinical practice, our study does not immediately show a benefit for EV enrichment prior to cfRNA isolation, especially in respect to time and cost effectiveness. In children, the amount of plasma is the major limiting step, and it seems simpler to just isolate RNA from full plasma than first performing density gradient centrifugation for EV isolation. However, that the concentration of EV can result in enrichment of tumor RNA was indeed recently shown by Stegmaier et al. [53]. They show that their target of interest, the transcripts of the *PAX-FOXO* or *SYT-SSX* fusion genes from alveolar rhabdomyosarcoma and synovial sarcoma, respectively, had a higher concentration in EV-derived cfRNA from patient plasma than cfRNA directly from plasma [53]. In future studies, it is important to determine which question needs to be answered. The purity of EV through elaborate isolation procedures can be essential to increase the knowledge of EV cargo and function, whereas a translational goal to improve diagnostic procedures might benefit more from a quick EV enrichment procedure through commercially available precipitating agents which increase the target concentration and thereby the sensitivity of the test.

## 5. Conclusions

In this study, we explore the possibilities of different cfRNA markers from plasma as novel biomarkers in patients with neuroblastoma. We discuss the possible variables affecting the detection of cfRNA-based markers and present approaches for correcting for the presence of platelets and background marker expression in plasma. Considering the neuroblastoma-specific markers, we conclude that these are only present in patients with metastatic disease. For the cell cycle markers, we find that many markers are higher in patients than in healthy controls, but only elevated above background expression levels in some metastatic patients. Our experiments on EV using SEC isolation illustrate that the mRNA markers are mostly expressed in EV-enriched SEC fractions, whereas cfDNA is mostly present in EV-poor SEC fractions. This study can form a starting point for further research into the potential of cfRNA-based analysis of liquid biopsies since this can be an additional approach to the more common analysis of cfDNA for liquid biopsies.

## Figures and Tables

**Figure 1 cancers-15-02108-f001:**
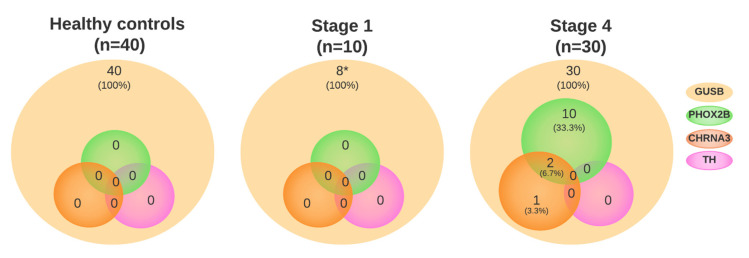
Expression of neuroblastoma-specific genes in cell-free RNA from healthy controls (*n* = 40), and diagnostic plasmas from patients with neuroblastoma with localized (*n* = 10) and metastatic (*n* = 30) disease. * Not enough material was left for 2 patients to perform the ddPCR for these neuroblastoma-specific markers.

**Figure 2 cancers-15-02108-f002:**
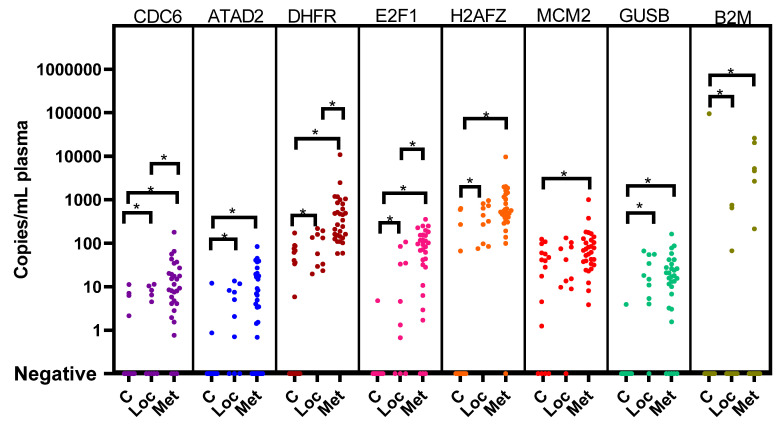
Expression of cell cycle genes (*CDC6*, *ATAD2*, *DHFR*, *E2F1*, *H2AFZ* and *MCM2*) and reference genes (*GUSB* and *B2M*) in cell-free RNA from healthy controls (*n* = 40) and diagnostic plasmas from patients with neuroblastoma with localized (*n* = 10) and metastatic (*n* = 30) disease, as measured by ddPCR from 200 µL plasma and corrected for platelet contamination. C; healthy controls. Loc; patients with localized disease. Met; patients with metastatic disease. * Significance at *p* < 0.05.

**Figure 3 cancers-15-02108-f003:**
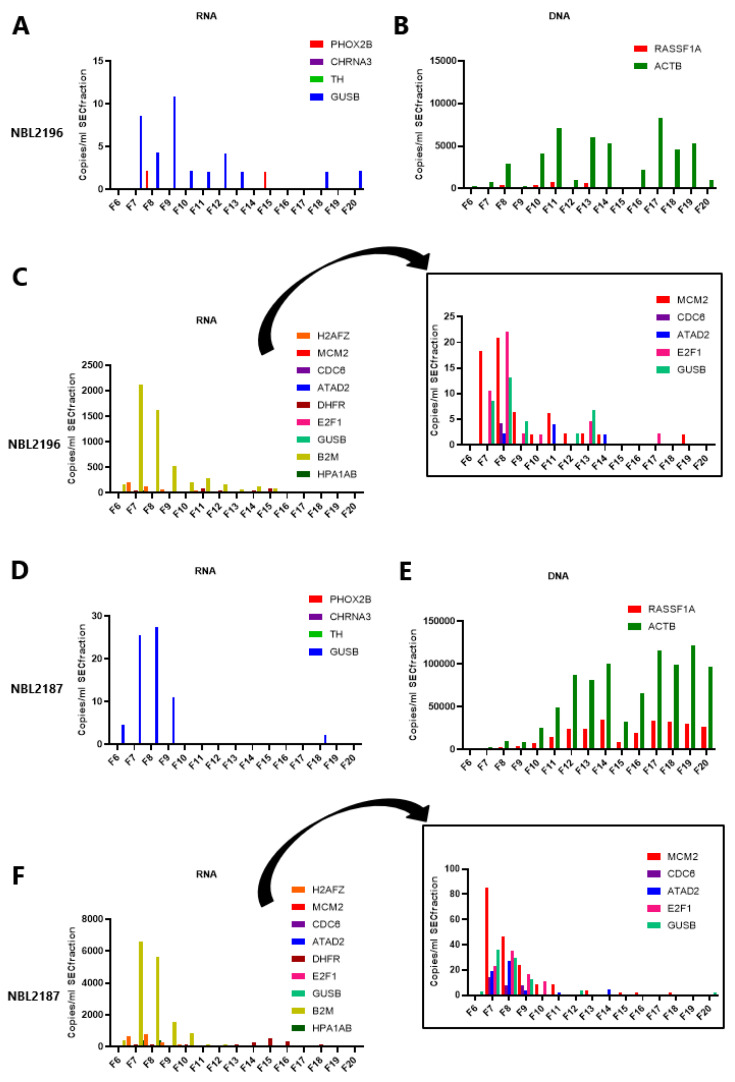
Expression of the mRNA markers and cell-free DNA markers in fractions isolated by size exclusion chromatography (SEC) and analyzed by ddPCR. Fractions F7 to F10 are considered as enriched in extracellular vesicles (EV). For patient NBL2196, (**A**) shows the expression of the neuroblastoma-specific mRNA markers, *PHOX2B*, *CHRNA3* and *TH*, and reference gene *GUSB* in cell-free RNA from 200 µL per SEC fraction: only *GUSB* and *PHOX2B* are expressed. (**B**) shows the cfDNA tumor-specific target methylated *RASSF1A* (*RASSF1A*-M) and reference gene *ACTB* in cfDNA from 200 µL per SEC fraction. (**C**) illustrates the expression of the cell cycle markers (*H2AFZ*, *MCM2*, *CDC6*, *ATAD2*, *DHFR*, *E2F1*, *GUSB*, *B2M* and *HPA1A/B*) in 200 µL per SEC fraction from the same patient. For patient NBL2187, (**D**) shows the expression of the neuroblastoma-specific mRNA markers, *PHOX2B*, *CHRNA3* and *TH*, and reference gene *GUSB* in cell-free RNA from 200 µL per SEC fraction: only *GUSB* is expressed. (**E**) shows the cfDNA tumor-specific target methylated *RASSF1A* (*RASSF1A*-M) and reference gene *ACTB* in cfDNA from 200 µL per SEC fraction. (**F**) illustrates the expression of the cell cycle markers (*H2AFZ*, *MCM2*, *CDC6*, *ATAD2*, *DHFR*, *E2F1*, *GUSB*, *B2M* and *HPA1A/B*) in 200 µL per SEC fraction from the same patient.

**Figure 4 cancers-15-02108-f004:**
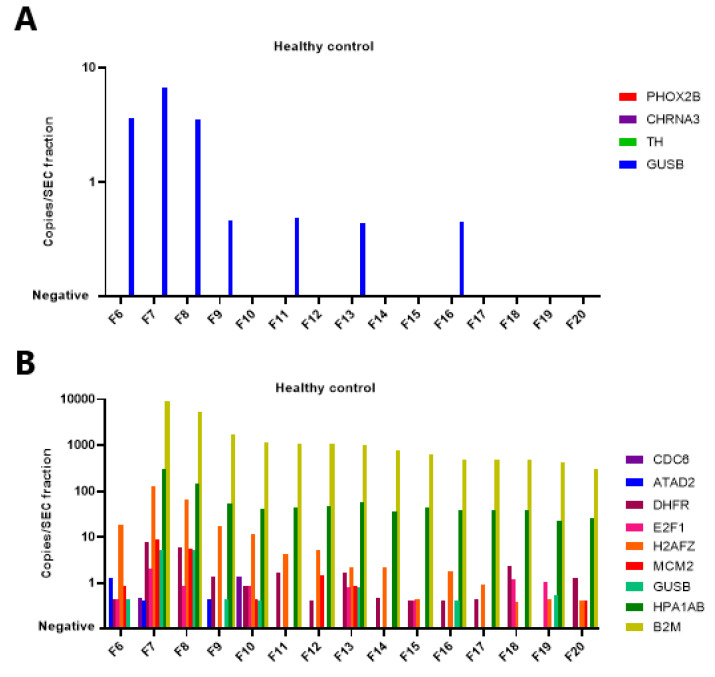
Expression of neuroblastoma-specific and cell cycle genes in 500 µL of SEC (size exclusion chromatography) fractions, as isolated from 500 µL plasma, from one healthy control and one patient with metastatic neuroblastoma. Fractions F7 to F10 are considered as enriched in extracellular vesicles (EV). In the healthy control, (**A**) shows the expression of the neuroblastoma-specific genes (*PHOX2B*, *CHRNA3* and *TH* with reference gene *GUSB*) and (**B**) shows the expression of the cell cycle markers (*CDC6*, *ATAD2*, *DHFR*, *E2F1*, *H2AFZ* and *MCM2* with *GUSB*, *HPA1A/B* and *B2M*). In the patient NBL 2177, (**C**) shows the expression of the neuroblastoma-specific genes (*PHOX2B*, *CHRNA3* and *TH* with reference gene *GUSB*) and (**D**) shows the expression of the cell cycle markers (*CDC6*, *ATAD2*, *DHFR*, *E2F1*, *H2AFZ* and *MCM2* with *GUSB*, *HPA1A/B* and *B2M*). Please note that the *y*-axis is represented on a log scale.

## Data Availability

The data presented in this study are available in this article (and Appendix A). Further data can be shared up on request.

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
