# Peer review of "Cell-Free RNA from Plasma in Patients with Neuroblastoma: Exploring the Technical and Clinical Potential"

_cancers, 2023, doi:10.3390/cancers15072108_

Round 1

Reviewer 1 Report

Overall, this is a high quality study with excellent supplementary material.  The only comment this reviewer would have is that the simple summary could be more simple.

Author Response

Reviewer 1

Quality of English Language

( ) English very difficult to understand/incomprehensible
( ) Extensive editing of English language and style required
( ) Moderate English changes required
(x) English language and style are fine/minor spell check required
( ) I am not qualified to assess the quality of English in this paper

Yes

Can be improved

Must be improved

Not applicable

Does the introduction provide sufficient background and include all relevant references?

(x)

( )

( )

( )

Are all the cited references relevant to the research?

(x)

( )

( )

( )

Is the research design appropriate?

(x)

( )

( )

( )

Are the methods adequately described?

(x)

( )

( )

( )

Are the results clearly presented?

(x)

( )

( )

( )

Are the conclusions supported by the results?

(x)

( )

( )

( )

Comments and Suggestions for Authors

Overall, this is a high quality study with excellent supplementary material.  The only comment this reviewer would have is that the simple summary could be more simple.

Authors: We are very grateful for the review of our manuscript and for the kind words. We have revised the simple summary to make it more simple:

Neuroblastoma mostly affects young children and despite intensive treatment, many children die of progressive disease. It remains challenging to identify those patients at risk. Analyzing blood, as liquid biopsies, is not invasive and can help to identify these patients. We studied whether RNA molecules can be detected in these liquid biopsies. In blood plasma, RNA can be free-floating or packed in small particles, ‘extracellular vesicles’. We present a workflow to analyze this cell-free RNA from small volumes of blood plasma of children with neuroblastoma. We have used neuroblastoma-specific markers and markers involved in cell proliferation. These latter genes can be upregulated in many different tumor types. We demonstrate that both types of markers have a higher expression in patients with metastatic disease, compared to healthy controls and patients with localized disease. These findings are essential for future studies on cell-free RNA, hopefully leading to improved survival for these patients.

Reviewer 2 Report

The paper entitled "Cell-free RNA from plasma in patients with neuroblastoma: exploring" described that isolated  cfRNA can be an interesting novel biomarker for detection of neuroblastoma, which has beedifficult to identify the patients at risk early during treatment. 

First, the final results were somewhat disappointed since the neuroblastoma-specific markers were present in plasma from 14/30 patients with metastatic disease. In other words, the sensitivity is only 46.7%. 

Second, these biomarkers were not present even in patients with localized disease. In other words, the selected biomarkers were not found in the localized disease but in the metastatic disease. How can these results be delineated?

Third, Authors should clearly describe the results without and with platelets. Also, why does it require to correct require the cfRNA results with platelet counts? 

As a result, authors extracted cfRNA form plasma including platelets, which could release lots of RNA upon activation. 

Unfortunately, autohrs did not describe how platelet-derived EVs and cfRNA were extracted. As authors may know, platelets are highly sensitive shear and yield lots of EVs and RNA, DNA, etc upon activation.

EV-derived RNA quantities are strongly dependent on the chosen method of EV isolation. This study adopted SEC. As reviewer understood, the recovery rate of the EVs would be very poor (leass than 20%) so that the biomarkers would be keep shaking whenever a biomarker discovery studies conducted. Thus, authors should reconsider the diagnostic performance of the present results which may be downgraded due to the EV isolation and cfRNA extraction methods. Our clinical studies found that the commercial products of EV isolation and DNA/RNA extraction were the main failure of the biomarker discovery study.

Specific comments

1. Fig. 1 is looging good but it is required to statistical results in graph format.

2. Authors used 200uL of plasma sample, which may not be enough to provide enough cfRNA for detection. 

3. Definition of cell cycle is required, even though authors may be comfortable to  use.

4.  cfRNA should be defined clearly whether it is directly from EVs or not. 

Author Response

Reviewer 2

Open Review

Quality of English Language

( ) English very difficult to understand/incomprehensible
( ) Extensive editing of English language and style required
( ) Moderate English changes required
( ) English language and style are fine/minor spell check required
(x) I am not qualified to assess the quality of English in this paper

Yes

Can be improved

Must be improved

Not applicable

Does the introduction provide sufficient background and include all relevant references?

(x)

( )

( )

( )

Are all the cited references relevant to the research?

( )

(x)

( )

( )

Is the research design appropriate?

( )

(x)

( )

( )

Are the methods adequately described?

( )

(x)

( )

( )

Are the results clearly presented?

( )

(x)

( )

( )

Are the conclusions supported by the results?

(x)

( )

( )

( )

Comments and Suggestions for Authors

The paper entitled "Cell-free RNA from plasma in patients with neuroblastoma: exploring" described that isolated  cfRNA can be an interesting novel biomarker for detection of neuroblastoma, which has beedifficult to identify the patients at risk early during treatment. 

 First, the final results were somewhat disappointed since the neuroblastoma-specific markers were present in plasma from 14/30 patients with metastatic disease. In other words, the sensitivity is only 46.7%. 
Second, these biomarkers were not present even in patients with localized disease. In other words, the selected biomarkers were not found in the localized disease but in the metastatic disease. How can these results be delineated?

Authors: We thank the reviewer for the careful review of our manuscript. The main aim of this study was exploring the potential for cell-free RNA analysis from low volumes of plasma, introducing multiplex ddPCR assays for cfRNA and solutions to challenges we encountered, e.g. platelet contamination of plasma. Initially, we were not sure if we would detect any markers, as we were investigating circulating cell-free RNA, which we assumed to be degraded because of the abundance of RNAses in our blood etc. Indeed, the neuroblastoma-specific markers were only detected in a proportionately low number of patients with metastatic disease and in none of the patients with localized disease. This difference might be explained by tumor characteristics. We would hypothesize that more aggressive tumor cells are more biologically active in shedding cellular components, e.g. EV, metabolites, cfRNA, proteins, cfDNA, than less disseminated and less aggressive tumors. Undoubtedly, this hypothesis requires further investigations.
Within this exploratory study, we did not want to claim that these neuroblastoma-specific markers bear any diagnostic or prognostic value if measured in the cell-free compartment, which is why we did not report sensitivity. It is not inconceivable that the neuroblastoma-specific markers that bear good prognostic value in the cellular compartment of the blood, as shown previously by our group and others, do not retain this value in the cell-free compartment. That is why we would argue that for use in the cell-free compartment, different markers must be sought and why we also explored the cell cycle markers. We added the following sentences to the start of the discussion:
Lines 331-333: This study addressed the potential for cfRNA analysis from small volumes of plasma using multiplex ddPCR assays and EV enrichment. Within this first exploratory study, we did not aim to draw conclusions on added value to current clinical practice.

And later in the discussion in lines 410-413: Specifically for patients with neuroblastoma, a study including RNA sequencing of the transcriptome of the cell-free compartment at different timepoints could improve understanding of this field immensely and help identify cfRNA markers with diagnostic and prognostic value. 

Third, Authors should clearly describe the results without and with platelets. Also, why does it require to correct require the cfRNA results with platelet counts? 
As a result, authors extracted cfRNA form plasma including platelets, which could release lots of RNA upon activation. 
Unfortunately, autohrs did not describe how platelet-derived EVs and cfRNA were extracted. As authors may know, platelets are highly sensitive shear and yield lots of EVs and RNA, DNA, etc upon activation.

Authors: Indeed, platelets are important variables in the analysis of cfRNA since they contain RNA themselves. Because the total platelet number were not available for the patient samples but the presence of platelets in the patient plasma would affect the levels of the different markers, we introduced the correction co-efficient for platelets. We would like to advocate that this platelet correction reflects the true cfRNA expression more realistically than without. We fully agree that we need to report both results with and without correction for platelets. These results were already present in Supplemental Table 9 but were not referred to correctly in the result section. We thank you for noticing and we have now added this as follow to the results section:
Lines 217-220: We found that platelets also contain these transcripts (Supplemental Figure 3 for expression in platelets). Since plasma was isolated by centrifugation at 1,375xg for 10 minutes, contaminating platelets might have been present in the plasma thereby affecting the analysis.

And in lines 233-234: … were significantly higher in metastatic patients than in localized patients (Figure 2 and Supplemental Table 9).

No EV were specifically isolated from platelets, only from total plasma. On the subject of platelet isolation from blood plasma to determine the background expression of the cell cycle markers, we have now added the following paragraph to the methods in lines 124-129:

Preparation of platelets from peripheral blood
Peripheral blood was collected in EDTA tubes (Becton-Dickinson) and processed within 2 hours. First, platelet-rich plasma was obtained by centrifugation at 235xg for 15 minutes. The supernatant was collected and 10% anticoagulant citrate dextrose, solution A (ACD-A) was added and centrifuged at
16 873xg for 4 minutes to pellet the platelets. Leukocyte and platelet counts were measured with the Sysmex XN1000 Hematology analyzer (Sysmex, Kobe, Japan) according to manufacturer’s protocol. 

EV-derived RNA quantities are strongly dependent on the chosen method of EV isolation. This study adopted SEC. As reviewer understood, the recovery rate of the EVs would be very poor (leass than 20%) so that the biomarkers would be keep shaking whenever a biomarker discovery studies conducted. Thus, authors should reconsider the diagnostic performance of the present results which may be downgraded due to the EV isolation and cfRNA extraction methods. Our clinical studies found that the commercial products of EV isolation and DNA/RNA extraction were the main failure of the biomarker discovery study.

Authors: We agree that the EV isolation method strongly affects the outcome. However, in our experience and as is also confirmed by the publication by van Deun et al. in 2014 (doi: 10.3402/jev.v3.24858) and recently by Vergauwen et al. (2021, doi: 10.1002/jev2.12122) EV isolation by SEC has proven to have good reproducibility and results in more than 70% recovery of the EV population. Still, since we know that the EV isolation method results in different EV yield, which can affect results, we standardized the method throughout all our experiments. We have now added the following to the discussion in line 422-425:
Furthermore, the method chosen for EV enrichment heavily affects result of the downstream analysis, and after SEC the presence of similar sized lipoprotein particles in the EV-enriched fractions might also result in less pure EV preparations. (50,51)

Specific comments

  1. 1 is looging good but it is required to statistical results in graph format.

Authors: We have added percentages to Figure 1, as shown below. As stated previously, this study was not aimed nor powered to draw any conclusions on the cfRNA analysis for diagnostic or prognostic value and therefore, we have not performed further statistical analysis.

  1. Authors used 200uL of plasma sample, which may not be enough to provide enough cfRNA for detection. 

Authors: We agree that 200uL is a very limited volume, however 200uL was the maximum volume available within this cohort for this study. On the other hand, we find that it supports the robustness of the techniques presented that even with these low volumes we were able to detect the different markers in the patient samples and even discern a distinct gene expression in patients with metastatic disease. Still, in a follow-up study, analysis of a larger volume of plasma would be preferrable to possibly broaden the markers (or perform RNA sequencing) and/or further studies on EV and other particles in the cell-free compartment.

  1. Definition of cell cycle is required, even though authors may be comfortable to  use.

Authors: We have now added the following sentence to the introduction in lines 94-96 and an additional reference:
Duplication of genomic DNA and distribution amongst the new daughter cells is a normal process in healthy cells. This process, named ‘cell cycle’, consists of well-defined phases, all guarded by checkpoints and their respective regulatory genes.(28)

  1. cfRNA should be defined clearly whether it is directly from EVs or not. 
    Authors: As stated previously, processing plasma with SEC results in fractions that are enriched in EV but these same fractions can also still contain other particles, e.g. lipoprotein particles, of the same size. Due to the potential presence of these particles, we cannot state that the cfRNA is purely from EV, so we refer to these as the ‘EV-enriched’ fractions. We have attempted to clarify this (as already stated in an answer above) with adding the follow to the discussion in lines 422-425:
    Furthermore, the method chosen for EV enrichment heavily affects result of the downstream analysis, and after SEC the presence of similar sized lipoprotein particles in the EV-enriched fractions might also result in less pure EV preparations. (51,52)

Moreover, we have added to both Figures 3 and 4 the following sentence to clarify which SEC fractions are enriched for EV:
Fractions F7 to F10 are considered as enriched in extracellular vesicles (EV).

Reviewer 3 Report

This manuscript deals with the detection of tumor-derived cell-free RNA (cfRNA) and tumor-specific extracellular vesicles from small volumes of blood plasma.

The authors designed several panels of markers to analyze both neuroblastoma-specific and cell cycle regulatory genes. They found these genes have a higher expression in patients with metastatic disease, compared to healthy controls and patients with localized disease.

They determined in a small fraction of these patients and a couple of healthy controls that some cfRNA were present in microvesicles.

The work is well performed, at least for the patients analyzed. The description of experiments is good, and the results are reliable and methods seems to be replicable. The significance and relevance of the findings reported is of interest, but actually it is not so evident the relevance of this present approach for the identification of the biological aggressiveness of neuroblastoma.

Actually, it is difficult to improve the manuscript, as I think that the data presented are the best of what the authors can show. The identification of cfRNA in microvesicles is of interest, but not all the patients analyzed have been assessed for the presence of cfRNA in these microvesicles.  A larger analysis of microvesicles from other patients is necessary to try to state something stronger than what is preliminarily supported.   

I would consider this manuscript preliminary, giving a message relevant but not fully demonstrated and mainly methodological.

Author Response

Reviewer 3

Open Review

Quality of English Language

( ) English very difficult to understand/incomprehensible
( ) Extensive editing of English language and style required
( ) Moderate English changes required
(x) English language and style are fine/minor spell check required
( ) I am not qualified to assess the quality of English in this paper

Yes

Can be improved

Must be improved

Not applicable

Does the introduction provide sufficient background and include all relevant references?

(x)

( )

( )

( )

Are all the cited references relevant to the research?

(x)

( )

( )

( )

Is the research design appropriate?

( )

( )

(x)

( )

Are the methods adequately described?

(x)

( )

( )

( )

Are the results clearly presented?

(x)

( )

( )

( )

Are the conclusions supported by the results?

( )

(x)

( )

( )

Comments and Suggestions for Authors

This manuscript deals with the detection of tumor-derived cell-free RNA (cfRNA) and tumor-specific extracellular vesicles from small volumes of blood plasma.

The authors designed several panels of markers to analyze both neuroblastoma-specific and cell cycle regulatory genes. They found these genes have a higher expression in patients with metastatic disease, compared to healthy controls and patients with localized disease.

They determined in a small fraction of these patients and a couple of healthy controls that some cfRNA were present in microvesicles.

The work is well performed, at least for the patients analyzed. The description of experiments is good, and the results are reliable and methods seems to be replicable. The significance and relevance of the findings reported is of interest, but actually it is not so evident the relevance of this present approach for the identification of the biological aggressiveness of neuroblastoma.

Actually, it is difficult to improve the manuscript, as I think that the data presented are the best of what the authors can show. The identification of cfRNA in microvesicles is of interest, but not all the patients analyzed have been assessed for the presence of cfRNA in these microvesicles.  A larger analysis of microvesicles from other patients is necessary to try to state something stronger than what is preliminarily supported.   

I would consider this manuscript preliminary, giving a message relevant but not fully demonstrated and mainly methodological.

Authors: We are grateful for the time invested in reviewing our manuscript. We fully agree that this manuscript is mainly methodological, demonstrating the feasibility of cfRNA analysis from small plasma volumes by multiplexed ddPCR and introducing solutions to common pre-analytical variables, e.g. platelet contamination of plasma. Indeed, further validation in a larger cohort is absolutely essential. Moreover, we would argue that the markers presented here might not be optimal and that additional markers should first be identified through sequencing efforts of the cell-free transcriptome compartment in this patient group. We have now further clarified our aim in the introduction as follows in the discussion:

Lines 331-334: This study addressed the potential for cfRNA analysis from small volumes of plasma using multiplex ddPCR assays and EV enrichment. Within this first exploratory study, we did not aim to draw conclusions on added value to current clinical practice.

 Lines 410-413: Specifically for patients with neuroblastoma, a study including RNA sequencing of the transcriptome of the cell-free compartment at different timepoints could improve understanding of this field immensely and help identify cfRNA markers with diagnostic and prognostic value. 

Round 2

Reviewer 2 Report

All questions were answered even though their quality was not enough. However, reviewer understood the limit of the present study.

Reviewer 3 Report

Actually, the Authors replied to my concerns, inserting some sentences in the manuscript. They did not perform additional experiments as expected. The manuscript is still preliminary, but I understand the point of view of the authors. Unfortunately, I cannot endorse the manuscript for publication.